# A Genome-Wide Association Study of 2304 Extreme Longevity Cases Identifies Novel Longevity Variants

**DOI:** 10.3390/ijms24010116

**Published:** 2022-12-21

**Authors:** Harold Bae, Anastasia Gurinovich, Tanya T. Karagiannis, Zeyuan Song, Anastasia Leshchyk, Mengze Li, Stacy L. Andersen, Konstantin Arbeev, Anatoliy Yashin, Joseph Zmuda, Ping An, Mary Feitosa, Cristina Giuliani, Claudio Franceschi, Paolo Garagnani, Jonas Mengel-From, Gil Atzmon, Nir Barzilai, Annibale Puca, Nicholas J. Schork, Thomas T. Perls, Paola Sebastiani

**Affiliations:** 1Biostatistics Program, College of Public Health and Human Sciences, Oregon State University, Corvallis, OR 97331, USA; 2Center for Quantitative Methods and Data Science, Institute for Clinical Research and Health Policy Studies, Tufts Medical Center, Boston, MA 02111, USA; 3Department of Biostatistics, Boston University School of Public Health, Boston, MA 02118, USA; 4Division of Computational Biomedicine, Boston University, Boston, MA 02215, USA; 5Chobanian & Avedisian School of Medicine, Boston University, Boston, MA 02215, USA; 6Social Science Research Institute, Duke University, Durham, NC 27708, USA; 7School of Public Health, University of Pittsburgh, Pittsburgh, PA 15260, USA; 8Department of Genetics, Washington University School of Medicine, St. Louis, MO 63110, USA; 9Department of Biological, Geological and Environmental Sciences, University of Bologna, 40126 Bologna, Italy; 10Department of Experimental, Diagnostic and Specialty Medicine, University of Bologna, 40126 Bologna, Italy; 11Department of Applied Mathematics and Laboratory of Systems Medicine of Aging, Lobachevsky University, 603950 Nizhny Novgorod, Russia; 12Department of Public Health, University of Southern Denmark, 5230 Odense, Denmark; 13Faculty of Natural Sciences, University of Haifa, Haifa 3498838, Israel; 14Department of Genetics and Medicine, Albert Einstein College of Medicine, Bronx, NY 10451, USA; 15Department of Medicine, Surgery and Dentistry “Scuola Medica Salernitana”, University of Salerno, 84084 Fisciano, Italy; 16Cardiovascular Research Unit, IRCCS MultiMedica, 20099 Milan, Italy; 17Quantitative Medicine & Systems Biology Division, Translational Genomics Research Institute, Phoenix, AZ 85004, USA

**Keywords:** human longevity, genetic variants, protein signatures

## Abstract

We performed a genome-wide association study (GWAS) of human extreme longevity (EL), defined as surviving past the 99th survival percentile, by aggregating data from four centenarian studies. The combined data included 2304 EL cases and 5879 controls. The analysis identified a locus in CDKN2B-AS1 (rs6475609, *p* = 7.13 × 10^−8^) that almost reached genome-wide significance and four additional loci that were suggestively significant. Among these, a novel rare variant (rs145265196) on chromosome 11 had much higher longevity allele frequencies in cases of Ashkenazi Jewish and Southern Italian ancestry compared to cases of other European ancestries. We also correlated EL-associated SNPs with serum proteins to link our findings to potential biological mechanisms that may be related to EL and are under genetic regulation. The findings from the proteomic analyses suggested that longevity-promoting alleles of significant genetic variants either provided EL cases with more youthful molecular profiles compared to controls or provided some form of protection from other illnesses, such as Alzheimer’s disease, and disease progressions.

## 1. Introduction

Multiple studies have presented evidence that exceptionally long-lived individuals are able to compress morbidity and disability towards the very end of their lives [1,2]. Together with this observation, our group has also shown that human extreme longevity (EL), defined as surviving past the 99th survival percentile, is a heritable trait, with increasing genetic influence as age approaches the extreme of human lifespan [3,4,5]. Therefore, centenarians provide a good model for examining healthy aging and studying the genetics of centenarians can lead to the identification of genetic factors that promote extreme health-span.

Many genome-wide association studies (GWASs) of EL have confirmed an association with the *APOE* locus [6]. However, to date, *APOE* remains the only genetic locus that has been associated with EL at the genome-wide significance level, with the association replicated across multiple cohorts. Other candidate loci have failed to reach the stringent genome-wide significance level [7,8], or replication in independent cohorts has failed. The yield of findings from GWASs of EL has not been commensurate with the yields of those for other complex genic traits, possibly due to the extreme rarity of the outcome and the difficulty in recruiting such individuals. Moreover, the results are dampened by the heterogeneity in genetic effects across different ethnicities [9,10] as well as the influence of multiple aging phenotypes, many with pronounced environmental effects [11,12].

In 2017, we conducted a meta-analysis of GWASs of EL that included the New England Centenarian Study (NECS), the Long Life Family Study (LLFS), the Southern Italian Centenarian Study (SICS), and the Longevity Genes Project (LGP) [8]. This analysis confirmed the association between EL and the *APOE* locus and discovered a few additional candidate loci for EL but lacked a replication study. In the current study, we conducted a GWAS of the data aggregated from the four centenarian studies and included 234 new EL cases to identify additional EL-associated genetic variants that are both common and rare. In addition to adding more cases, we used a different imputation reference panel, the Haplotype Reference Consortium (HRC), which was shown to have improved accuracy, especially for rare variants [13]. We also applied saddle point approximation (SPA) to the obtained score tests to yield more accurate test results for both common and rare variants [14,15]. We sought the replication of our discovery results in three publicly available GWASs of parental lifespans/survival, including the UK Biobank (UKB) [16] GWAS of father’s age at death and mother’s age at death and the meta-analysis of parental lifespan from the UKB and 26 independent European-heritage population cohorts (UKB+LifeGen) [17]. Finally, we used serum proteomic data in the NECS to prioritize possible longevity variants and link our findings to biological pathways important for longevity.

## 2. Materials and Methods

### 2.1. Study Populations and Genetic Data

#### Longevity Studies

This consortium included four studies of longevity with genome-wide genotype data. The studies and the selection of additional controls were previously described in reference [7]. The aggregated set included 2304 EL cases and 5879 controls. The NECS contributed 1296 cases (median age = 104 years, age range = (97, 119) years). The LLFS contributed 569 cases (median age = 101 years, age range = (97, 111) years). The LGP contributed 313 cases (median age = 102 years, age range = (96, 115) years). The SICS contributed 126 cases (median age = 99 years, age range = (96, 108) years). We imputed genome-wide genotype data in each study to the HRC panel (version r1.1 2016) of 64,940 haplotypes with 39,635,008 sites using the Michigan Imputation Server [18]. We analyzed approximately 1.4 million genotyped and imputed SNPs that passed an imputation quality score threshold of 0.7, a Hardy–Weinberg Equilibrium *p*-value threshold of 10^−6^, and additional stringent quality-control steps (see Appendix A for details) and that had a minor allele count (MAC) of 3 or more for both cases and controls.

### 2.2. Definition of Extreme Longevity Phenotype

We defined extreme longevity as an individual’s surviving beyond the 99th survival percentile in their sex and birth-year cohort (males: 96 years for 1900, 97 years for 1910, 98 years for 1920; females: 100 years) based on the US social security administration cohort tables [19]. We used this definition of EL in the mega-analysis of the four longevity studies.

We defined the controls as study participants who did not achieve the above threshold or study controls. In the NECS, study controls were defined as NECS referent subjects who were spouses of centenarian offspring or children of individuals who died at an age ≤ 73 years and matched the life expectancy of their birth cohort. In the LLFS, study controls were defined as spouses of members of the family selected for longevity. In the LGP, study controls were defined as genetically matched offspring of parents with usual survival (i.e., both parents died before the age of 85). In the SICS, study controls (age range = 18–48 years) were recruited from an isolated region of Southern Italy east of Naples with a high prevalence of longevity and health and characterized by a high level of endogamy. To increase statistical power, we also included additional controls from the Illumina repository as in prior studies of longevity with NECS and LLFS data [7,8]. This repository included approximately 6000 samples of various races and ethnicities used as controls for a variety of genome-wide association studies. Through a series of principal component analyses, we selected a sub-sample of these controls that matched the ethnic composition of the NECS and LLFS. Ages of death for some of these controls were unknown, but since we expected that only a very small portion of them would live to extreme old ages, we included all of them to avoid selection bias and bias against the null.

### 2.3. Replication Cohorts

#### 2.3.1. UKB Father and Mother

We downloaded the summary statistics for the GWASs of father’s age at death (UKB-F) and mother’s age at death (UKB-M) from the Pan-UK Biobank [20] (https://pan.ukbb.broadinstitute.org/, accessed on 31 May 2022), which houses summary statistics from multi-ancestry analysis of 7228 phenotypes, across 6 continental ancestry groups, for a total of 16,131 genome-wide association studies. For father’s age at death, 310,232 participants of European ancestry were included in the analysis. For mother’s age at death, 249,247 participants of European ancestry were included.

#### 2.3.2. UKB+LifeGen

UKB+LifeGen performed a large-scale GWAS of parental survival combining data from parents of European ancestry in the UKB and a previously published meta-analysis of 26 additional independent European-heritage population cohorts, totaling 1,012,240 parents. In the Lifegen+UKB cohort, UKB contributed 259,003 paternal ages at death with 80,729 censored observations and 210,609 maternal ages at death with 141,280 censored observations. The LifeGen consortium contributed 77,163 paternal ages at death with 83,298 censored observations and 62,364 maternal ages at death with 97,794 censored observations. The investigators examined the association between participants’ genotypes and parental survival using a residualized Cox model and Martingale residuals to transform survival into a quantitative trait. In the UKB, a sex-stratified analysis was performed and then the allelic effects in relation to paternal and maternal survival were combined into a single parental survival effect. The results for parental survival in the UKB and the meta-analysis of 26 cohorts in LifeGen were then meta-analyzed using inverse variance weighting. Detailed information about the cohorts and analysis plans can be found in reference [17].

### 2.4. Statistical Analysis

We combined the data from the four centenarian studies into one data set, and we tested the association between each genetic variant and EL using a mixed-effects logistic regression model adjusted by sex, the first four principal components, an indicator variable for residence in Southern Italy, an indicator variable for residence in Denmark, and the full genetic relationship matrix (GRM). Prior to the association testing, we removed participants of non-European ancestry based on a visual inspection of their principal component values. We used saddle point approximation (SPA) [15] of the derived score statistics to calibrate *p*-values [21]. A *p*-value < 5 × 10^−8^ was used as the genome-wide significance level, and *p* < 5 × 10^−6^ was used as a suggestive level of significance. We used our GWAS pipeline tool developed by Song et al. [22] to calculate the genome-wide principal components, the GRM, as well as the association testing and SPA-based *p*-values that we recently validated against other programs [21] (see the Appendix A for an overview of the pipeline). The results are displayed in Table 1.

### 2.5. Replication Criteria

Please note that we analyzed the associations between SNPs and the EL phenotype in the four longevity studies using logistic regression, while a censored survival analysis of parental lifespan of the enrolled offspring was used in the replication cohorts. Therefore, a significant variant in the discovery GWAS was replicated in the replication cohorts if the same variant had a consistent direction of effect and a nominal *p*-value < 0.05 in the replication cohort. For example, an allele’s increasing the odds of being an EL case in the discovery GWAS corresponded to an effect that increased parental lifespan in the replication cohort.

### 2.6. Protein Quantitative Trait Loci (pQTL) Analysis

We excluded from this analysis the APOE locus that we analyzed previously using the same data [23] and we focused attention on the four lead SNPs on chromosomes 4, 5, 9, and 11 (Table 1) that were associated with EL at *p* < 5 × 10^−6^ in the discovery GWAS and correlated these SNPs with serum proteins in the NECS (n = 220). The SNP rs145265196 on chromosome 11 was rare with a minor allele frequency (MAF) of 0.003, and there was only 1 carrier of the longevity allele in the proteomic data. Therefore, this SNP was excluded from the pQTL analysis. We used serum proteomics data of 220 NECS participants that we generated using the Somalogic aptamer-based technology, as described in [23]. The serum proteomic data included 4785 aptamers targeting 4116 unique human proteins that passed a quality-control assessment for median intra- and inter-assay variability. Log-transformed values of protein expressions were regressed on each of the three lead SNPs, adjusting for age at blood draw and sex. To account for the non-independence of 4785 aptamers, we estimated the effective number of independent proteins by applying the method proposed in [24]. We determined that the effective number of independent proteins was 60, which explained >99% of variability in the entire proteomic data. Based on this, a proteome-wide significance threshold of 0.05/60 = 0.00083 was used to identify the protein signatures for three SNPs. 

**Table 1 ijms-24-00116-t001:** Summary of Lead SNPs in Significant Loci.

				Discovery GWAS						UKB-F			UKB-M			UKB+LifeGen		
rsID	Gene	Chr	Pos	EA/NEA	EAF in Cases	EAF in Controls	Beta	SE	*p*	Beta	SE	*p*	Beta	SE	*p*	Beta	SE	*p*
rs429358	APOE	19	45411941	T/C	0.95	0.88	0.84	0.065	1.94 × 10^−36^	0.020	0.0034	3.27 × 10^−9^	0.019	0.0036	2.58 × 10^−7^	0.106	0.0055	3.14 × 10^−83^
rs6475609	CDKN2B-AS1	9	22106271	A/G	0.49	0.42	0.21	0.039	7.13 × 10^−8^	0.019	0.0025	1.41 × 10^−14^	0.006	0.0027	0.03	0.024	0.0039	9.98 × 10^−10^
rs145265196	RPLP0P2	11	61401362	G/T	0.007	0.002	1.74	0.347	6.29 × 10^−7^	−0.022	0.0405	0.59	0.025	0.0443	0.57	NA	NA	NA
rs9657521	OR7E161P| DEFB136	8	11830502	A/C	0.76	0.71	0.20	0.044	3.86 × 10^−6^	0.009	0.0027	0.0012	0.005	0.0029	0.07	0.013	0.0043	0.0021
rs145282854 *	ZBED1P1| ENPEP	4	111244992	A/G	0.022	0.013	0.72	0.157	5.47 × 10^−6^	−0.013	0.0124	0.29	−0.014	0.0134	0.30	0.003	0.0195	0.89

EA = Effect (coded) allele (the longevity-promoting allele), NEA = Non-effect allele, EAF = Effect allele frequency, Beta = log odds ratio for EL associated with each additional effect allele, SE = standard error of beta. * Did not reach suggestive significance (*p* < 5 × 10^−6^).

### 2.7. Gene Set Enrichment Analysis

We performed a gene set enrichment analysis of the proteomic signatures using the human HALLMARK gene set compendium and the Gene Ontology (GO) Biological Processes, Cellular Components, and Molecular Functions retrieved from msigDB [25]. We conducted the enrichment analysis using the hypeR [26] R package, with the hypergeometric test and the overlap between all the genes in each compendium and the whole list of proteins analyzed in our analysis as background.

### 2.8. Phenome-Wide Association Study (PheWAS) Search

We also conducted a regional phenome-wide association study (PheWAS) [27] search of the associations between the top SNPs with 778 traits in 30 million genetic variants computed with 452,264 UK Biobank White British individuals (http://geneatlas.roslin.ed.ac.uk/, accessed on 18 September 2022) to potentially link the identified variants to other traits/diseases that may be relevant to EL.

## 3. Results

A flow chart that illustrates the study design is shown in Figure 1. The Manhattan plots in Figure 2 summarizes the results of the GWAS. We decided to focus attention on the five loci that were associated with EL at *p* < 5 × 10^−6^ and were either replicated in the independent sets or were rare variants that were more frequent in centenarians and for which the associations were supported by a cluster of SNPs in linkage disequilibrium. Table 1 includes the results of the lead SNPs in these five loci, and the complete set of GWAS results, with *p* < 5 × 10^−6^, along with the replication results, can be found in Appendix A. 

The associations between EL and a cluster of 30 SNPs in the *APOE* locus (top SNP: rs429358, *p* = 1.94 × 10^−36^) reached genome-wide significance and were replicated in the UKB-F, UKB-M, and UKB+LifeGen. The association between the E2 allele of *APOE* and EL is well established [28], and we previously determined a serum proteomic signature of the *APOE* alleles that has been well replicated [23,29]. Therefore, we will describe in detail the associations of the other loci.

### 3.1. Locus on Chromosome 9: CDKN2B-AS1

A cluster of 36 SNPs in the long noncoding RNA (lncRNA) gene *CDKN2B-AS1* had *p*-values < 5 × 10^−6^, and the lead SNP rs6475609 fell slightly short of the genome-wide significance with *p* = 7.13 × 10^−8^ and was replicated in all three replication cohorts. Interestingly, the effect size of the association between rs6475609 and parental survival was much stronger in the UKB-F (beta = 0.019, *p*= 1.41 × 10^−14^) than in the UKB-M (beta = 0.0057, *p* = 0.03), suggesting a possible sex-specific effect. Thus, we examined this SNP separately for males and females in our discovery data using the same model used for the genome-wide analysis and confirmed the same trend with beta = 0.016 (*p* = 0.05) in males and beta = 0.026 (*p* = 6.74 × 10^−6^) in females. The SNP rs6475609 is a common intronic variant in gene *CDKN2B-AS1* and was previously found to be associated with EL, although the association did not reach a genome-wide level of statistical significance [29,30]. When we correlated the SNP rs6475609 with serum proteomic data (pQTL), we found a signature of nine aptamers mapping to eight proteins: C-C Motif Chemokine Ligand 15 (CCL15), Chromogranin A (CHGA), Kallikrein Related Peptidase 10 (KLK10), Mitochondrial Fission Factor (MFF), Pro-Platelet Basic Protein (PPBP), LDL Receptor Related Protein 11 (LRP11), Quiescin Sulfhydryl Oxidase 2 (QSOX2), and Zinc And Ring Finger 3 (ZNRF3) (Figure 3 and Table 2). Although this signature was not enriched for any pathway of the gene set, we noticed that five of these eight proteins (CCL15, CHGA, KLK10, LRP11, and PPBP) were associated with age at 1% FDR in the analysis we published in [31] (see columns FC and AdjP in Table 2). The current analysis showed that individuals carrying the longevity allele A of rs6475609 had lower expression of CCL15 (consistent for both aptamers), CHGA, and KLK10 and that the abundances of these three proteins increased with age. The protein KLK10 also replicated its association with genetic variants in the *CDKN2B-AS1* gene with the same trend, as found in our previous analyses [29]. Conversely, individuals carrying the longevity allele had higher expression of PPBP that decreased with older age. We observed this trend previously and noted that carriers of the longevity allele of *CDKN2B-AS1* had younger profiles for these and other aging biomarkers that are maintained throughout the lifespan.

### 3.2. Locus on Chromosome 11: RPLPOP2

We observed a suggestively significant, although not genome-wide significant, peak on chromosome 11 that harbors 25 rare SNPs in *RPLPOP2* with *p* < 5 × 10^−6^ in our discovery GWAS. The association of the lead SNP rs145265196 reached a *p* = 6.29 × 10^−7^ level of significance, although this association failed to be replicated in the UKB-F and UKB-M and this SNP was not available in the UKB+LifeGen study that focused on variants with minor allele frequencies > 0.005. The MAF of this SNP in the EL cases was 0.0067 and in the controls was 0.0015, which roughly represents a 4.5-fold enrichment. The frequency of this allele was 0.00094 in the UKB-M and 0.00095 in the UKB-F—lower than the range of MAFs reported in TopMed (0.002) and gnomAD (0.001) and ALFA (0.001). We only found one carrier of the longevity allele in the proteomic data set and could not perform the pQTL analysis.

### 3.3. Locus on Chromosome 8

On chromosome 8, we observed a stretch of 18 variants with *p*-values < 10^−4^ (top SNP: rs9657521, *p* = 3.86 × 10^−6^). The association of the lead SNP was replicated in the UKB-F and UKB+LifeGen with a consistent direction of effects, and the association reached statistical significance after correction for three tests (*p* < 0.017) in the UKB-F and UKB+LifeGen. However, it failed to be replicated in the UKB-M (*p* = 0.067), although it had a consistent direction of effects. This locus is in an intergenic region between *OR7E161P* and *DEFB136* on 8p23.1, which harbors genes such as *GATA4*, *NEIL2*, *FDFT1*, *CTSB*, and *DEFB136*. The top SNP rs9657521 is found downstream of *DEFB136*. Annotations from the pQTL results (Figure 4) detected one protein SLAM Family Member 6 (SLAMF6) as being significantly associated with rs9657521, after correction for multiple testing, and three additional proteins (Interleukin 18 Binding Protein (IL18BP), *p* = 0.000862; Proprotein Convertase Subtilisin/Kexin Type 1 Inhibitor (PCSK1N), *p* = 0.000929; and Ciliary Neurotrophic Factor Receptor (CNTFR), *p* = 0.000949) that barely missed the proteome-wide significance level of 0.00083. SLAMF6, IL18BP, and CNTFR were associated with age at 1% FDR in the analysis we published in reference [31]. The signature of four proteins was not enriched for any biological pathways, but we noted that carriers of the longevity allele had higher abundances of the three biomarkers that increase with age.

### 3.4. Locus on Chromosome 4

The association of an uncommon variant rs145282854 in *ZBED1P1*|*ENPEP* reached a 5.47 × 10^−6^ level of significance, which barely missed the suggestive significance. This SNP failed to be replicated in all three replication sets. This SNP is an uncommon variant, for which the longevity allele frequencies in cases and controls were 0.022 and 0.013, respectively. The allele frequency of this SNP was 0.01 in the UKB-M and UKB-F and 0.00875 in ALFA. Although the evidence for genetic association was weak, the correlation with proteomic data showed that rs145282854 was associated with a signature of 14 proteins, 9 of which were associated with age at 5% FDR (Table 2, Figure 5). The signature included tumor necrosis factor 15 (TNFS15), which decreases with older age, and carriers of the longevity variant had lower expression levels compared with non-carriers.

## 4. Discussion

We conducted a GWAS of EL by aggregating the individual-level data from four longevity studies, making for a total of 2304 EL cases and 5879 controls. In addition to confirming the association with *APOE*, we found additional loci that were suggestively significant and replicated in independent studies. Unlike common diseases or phenotypes, a GWAS of extremely rare phenotypes, such as EL, is still underpowered when it comes to detecting true associations at the stringent genome-wide significance level. Therefore, we also correlated EL-associated SNPs with serum proteins to cast light on potential biological mechanisms that are related to EL and under genetic regulation.

With the inclusion of additional EL cases, *CDKN2B-AS1* variants nearly achieved genome-wide significance using EL as a trait, instead of utilizing the reported parental lifespans in other studies. In a prior analysis conducted by our group [29], the top variant (rs2184061; *p* = 3.82 × 10^−7^) in *CDKN2B-AS1* fell short of the genome-wide significance level. In the current study, this locus almost achieved genome-wide significance (rs6475609, *p* = 7.13 × 10^−8^). The SNP rs6475609 was associated with a signature of eight serum circulating proteins that includes five aging biomarkers. Three of these proteins—CCL15, CHGA, and KLK10— are known to be prognostic markers for various types of cancer, and upregulation of these proteins is correlated with poorer prognosis [32,33,34]. LRP11 is predicted to act in response to many biological processes linked to Alzheimer’s disease (AD) [35]. Consistent with the analysis reported in [29], our analysis showed that expression levels for all four proteins increased with old age, but carriers of the longevity allele had lower levels of these proteins in serum than carriers of the non-longevity allele. PPBP stimulates a variety of processes, including activation of neutrophils, which is the immune system’s first line of defense [36]. The abundance of this circulating protein declines with older age, possibly marking immune system exhaustion. However, carriers of the longevity allele appear to maintain higher values of this biomarker across different ages (Figure 3). The relations between age, the longevity allele, and protein abundance suggest that the longevity variant of *CDKN2B-AS1* may help individuals maintain more youthful profiles of these biomarkers as they age. In addition, the protein MFF was higher in carriers of the longevity variant. Mitochondrial fission is an essential process for the removal of defective mitochondria through various mechanisms, such as mitophagy, mitochondrial transport, and programmed cell death [37]. Evidence from model organisms has shown that increasing mitochondrial fission (i.e., higher levels of MFF) and mitophagy in middle-aged animals correlates with longer lifespan. Our analysis suggests that having the longevity variant helps sustain the ability to execute appropriate mitochondrial fission at old ages.

We discovered a suggestively significant (although not genome-wide significant) locus on chromosome 8 (rs9657521) that was associated with EL and which has not previously been reported in the literature. This locus is in a region of 8p23.1, which contains the genes *GATA4*, *NEIL2*, *FDFT1*, *CTSB*, and *DEFB136*. The protein signature that we found to be associated with rs9657521 includes PCSK1N, also known as proSAAS, IL18BP, SLAMF6, and CNTFR. The longevity-promoting allele of rs9657521 was associated with lower levels of the serum protein PCSK1N (see Figure 4). The gene PCSK1N is widely expressed in the brain, and its expression increases in the brains of rodents subjected to hypoxia and dehydration [38]. It has been identified as a cerebrospinal fluid candidate biomarker for AD and/or dementia [38], and a recent transcriptomic analysis showed that PCSK1N expression increased during AD progression [39,40]. Our results suggest that the longevity allele of the SNP rs9657521 helps maintain lower values of this protein in the serum and can mark protection from AD or some other mechanisms that need further investigation. In addition, protein abundance of IL18BP appeared to increase with older age, and carriers of the longevity variant had higher expression levels compared to carriers of the non-longevity variant. The effect of IL18BP is to reduce interleukin 18 activity, which is a pro-inflammatory protein involved in a variety of processes that can lead to organ injury and possibly a fatal condition characterized by cytokine storms. For example, higher levels of IL18 were markers of poor prognosis in COVID-19 patients [41]. Therefore, higher values of IL18BP should correlate with lower values of IL18 and their increase with older age is likely to represent a protective mechanism against inflammation that is enhanced in carriers of the longevity variants. 

In the literature, rs9657521 has been associated with the expression of the gene cathepsin B (*CTSB)* in blood (Open Target Genetics Portal [42]). *CTSB* was one of 42 newly discovered loci in a recent GWAS meta-analysis of AD [43]. *CTSB* was also shown to be linked to Parkinson’s disease [43,44]. Among the top associated traits in the PheWAS search, BMI, platelet distribution width, and red blood cell distribution width were negatively correlated with the longevity allele of this SNP (*p* < 1 × 10^−12^). Higher values of these traits are strongly predictive of mortality, incident coronary heart disease, and cancer [45], and it is interesting that carrying the longevity allele appears to confer protection. Our analyses suggest that this is an important locus for longevity, but additional replication of the genetic and molecular associations is needed to pinpoint the exact biological mechanism by which this locus influences EL. 

The SNP rs145265196 is an intronic rare variant in *RPLPOP2*. The aggregated data of four longevity studies in the current analysis allowed for a more careful examination of a comparison of allele frequencies (AFs) by distinct ethnicities that included individuals of Danish, Ashkenazi Jewish, Southern Italian, and Central European ancestries. Individuals in the “central” European ancestry group were individuals who did not belong to any of the three distinct ethnic groups (Danish, Ashkenazi Jewish, and Southern Italian) but who formed a cluster of their own. Among the Southern Italian individuals, the cases had much higher AFs (0.012) compared to controls (0.0023). Similarly, among the Ashkenazi Jewish individuals, cases had a longevity AF of 0.012 compared to 0.0024 in controls. Similar but somewhat weak trends were observed in individuals of Danish ancestry (case AF = 0.005 vs. control AF = 0.0022) and Central European ancestry (case AF = 0.004 vs. control AF = 0.0008). This examination revealed that the longevity allele of this rare variant was much more prevalent among the cases of Southern Italian and Ashkenazi Jewish ancestries and confirmed that there exists ethnicity-dependent heterogeneity in the association between EL and genetic variants [10]. Hence, this presents a potential future avenue for investigating the genetic effects on EL, as recently supported in Giuliani et al. 2018 and other studies [9,11,46].

SUMO Specific Peptidase 7 (SENP7) and CCCTC-Binding Factor (CTCF), which are part of the signature for rs145282854 on chromosome 4, also presented interesting examples. In our analysis, carriers of the longevity allele had higher expression levels of these protein compared to non-carriers. Findings from a recent study revealed that SENP7 may act as an oxidative stress sensor to maintain metabolic fitness and antitumor functions in CD8+ T cells [47]. Therefore, the effect of the longevity variant may be to provide adequate stress sensing. In a different study that examined functional roles of SENP7 [48], the authors concluded that proper neuronal differentiation requires SENP7 and that SENP7 could be a key regulator in neuronal differentiation. Moreover, given that neurogenesis has been shown to be impaired at early stages of AD [49], the role of SENP7 may be potentially important for pathways that influence AD progression. There has also been emerging evidence that CTCF may play a vital role in DNA damage response by facilitating DNA double-strand break repair [50]. Thus, our results may suggest that the longevity variant may help provide DNA-repair mechanisms at older ages, which has also been found in a whole-genome sequencing analysis of semi-supercentenarians in Italy [51].

The selection of appropriate controls presents a challenge in genetic studies of longevity. An ideal set of controls for longevity studies would be individuals who were born in the same birth-year cohort as the cases and who had usual survival. However, obtaining DNA samples for these birth-year matched controls is nearly impossible in centenarian studies. In our study, with the exception of the SICS, the study controls were selected from the general population with no evidence of longevity, so there is likely very little selection bias. Additionally, most of the study controls are still alive and could eventually become centenarians later. Our rationale was that the prevalence of centenarians is very rare, so that any individual selected from the general population will have a very low chance of becoming a centenarian in the future, and that inclusion of some centenarians in the control set would lower statistical power but not introduce biases [52]. We also acknowledge that the effects derived from our study may be biased; the effect estimates we obtained may be larger or smaller than the actual effects due to some level of selection bias. Nonetheless, the goal of this analysis was hypothesis testing, not estimation, to identify genetic variants for which the distribution of alleles was different between the cases and controls.

Ensuring high imputation quality, especially for rare variants, is a crucial task to avoid spurious associations in GWASs. We used an imputation quality score > 0.7, given that a score > 0.5 was used in the original article describing the reference panel [13] and that a score > 0.3 is also commonly used [53]. Additionally, the authors of reference [13] noted increases in the imputation quality with the HRC panel in comparison with the 1000 Genomes Project (1000GP) panel with imputation quality R^2^ = 0.64 (HRC) vs. R^2^ = 0.36 (1000GP) at MAF = 0.1%. Therefore, we believe that the threshold of 0.7 was reasonable. We further investigated the quality of an imputed rare variant (rs145265196) on chromosome 11 by comparing the imputed dosage data with the whole-genome sequence data in the LLFS. There were 4241 overlapping LLFS subjects who had both imputed data and whole-genome sequence data. For these participants, the computed MAC using the imputed data was 52.798, and the MAC using the whole-genome sequence data was 52, which resulted in a correlation coefficient of 0.9989. Therefore, we believe that the quality of imputation for this SNP was very good. Moreover, in a paper that was recently accepted for publication [54,55,56], our group performed the same type of comparison between the imputed data and whole-genome sequence data for the top rare SNPs with an MAF of 0.0005 and we observed a perfect concordance. Therefore, these rare variants imputed to the HRC panel with high quality appear to be trustworthy. 

In addition to better imputation of uncommon and rare variants, we also used improved modeling techniques to test the associations between SNPs and EL using logistic regression. Compared with past analyses [8], we adopted a mixed-effects logistic regression model using a full GRM and SPA of the score test that impacted the calculations for some of the suggestive associations between chromosomes 4, 7, and 12. For example, the level of significance for rs28391193 on chromosome 4 changed from 2 × 10^−7^ in the meta-analysis to 1 × 10^−4^ in the current analysis. 

There are a few limitations to this study. First, the sample size for the NECS proteomic data was 220, which may have led to low power in detecting significant associations. It is possible that some nominally significant results were true-positive associations which we failed to detect. Second, our results from the proteomic analysis did not have a replication set. Thus, the protein signatures we identified remain to be replicated in a future study. We acknowledge that additional multi-omics data on a larger number of participants and the incorporation of other approaches, such as haplotype-based methods, could help elucidate the potential biological mechanism for identified loci in a future analysis. Third, our replication cohorts relied on a censored survival analysis of offspring report of parental lifespan that was not verified, while a logistic model was used for the EL phenotypes of enrolled participants in the discovery cohort. Additionally, the inherent parent–offspring design may have introduced unnecessary noise into the replication results. Fourth, the current analysis, which was restricted to participants of European ancestries, may not be generalizable to participants of other non-European ancestries. Of note, we also showed that the genetic effects can vary even among individuals of different European ancestries for the rare SNP rs145265196. Lastly, the three replication cohorts that used the data from the UKB were not independent. Still, we believe that it was important to note any sex-specific effects of longevity variants. 

## 5. Conclusions

Genetic studies of EL have been challenged by limited sample sizes that have made it extremely difficult to reach genome-wide levels of significance for many putative associations with limited effects. In this study, we tried to circumvent the limited sample sizes of individual studies by aggregating the data from four centenarian studies into one larger analysis that provided greater power to detect suggestive associations of rare and uncommon variants. Still, our sample size was limited, and although we do not have formal ways to calculate the proportion of genetic variability explained by this set of variants, we posit that these new loci explain a small portion of the genetics of EL, while much more remains to be found. Although replication of these results in other studies is warranted, the analysis showed new interesting variants associated with EL. The integration of genetic data with serum proteomic data also pointed to potential interesting molecular processes that are under genetic regulation and which may be implicated in various pathways related to living to extreme old age. Such mechanisms may provide new targets for healthy aging therapeutics. 

## Figures and Tables

**Figure 1 ijms-24-00116-f001:**
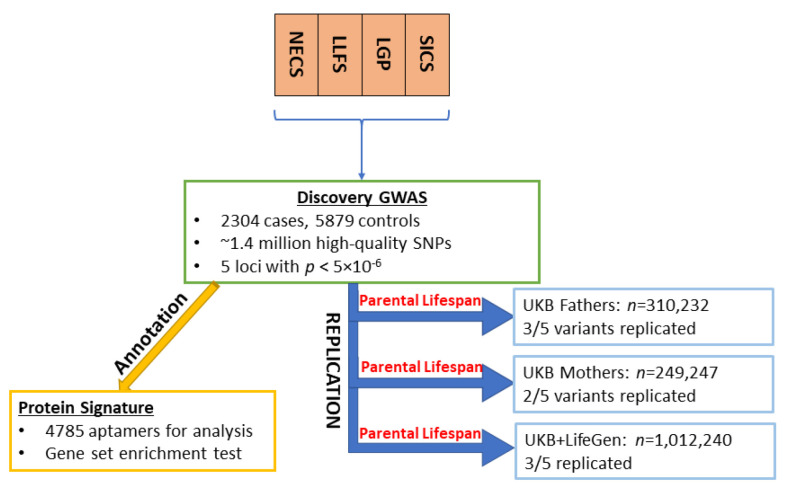
Flow chart of the study design.

**Figure 2 ijms-24-00116-f002:**
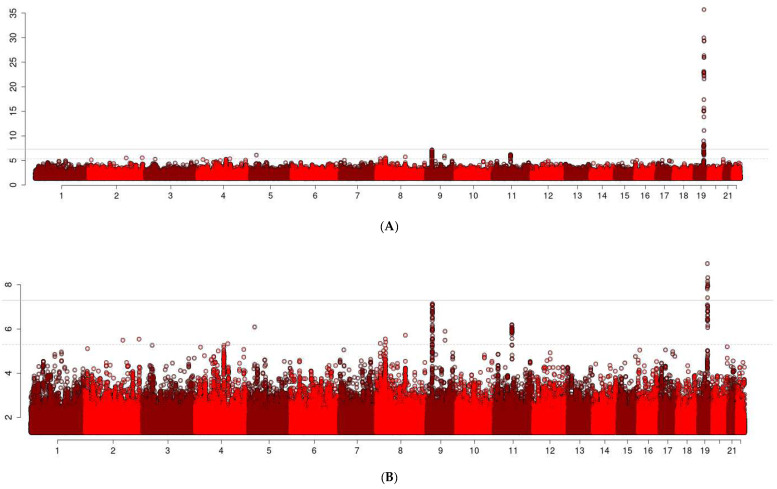
Manhattan Plot of the Discovery GWAS. The *x*-axis reports chromosomes and coordinates within chromosomes. The *y*-axis reports the −log10 of *p*-values. Top Panel (**A**) shows the unscaled Manhattan plot. Bottom Panel (**B**) shows the truncated version, where the *y*-axis shows only up to 10 (i.e., *p*-value of 10^−10^).

**Figure 3 ijms-24-00116-f003:**
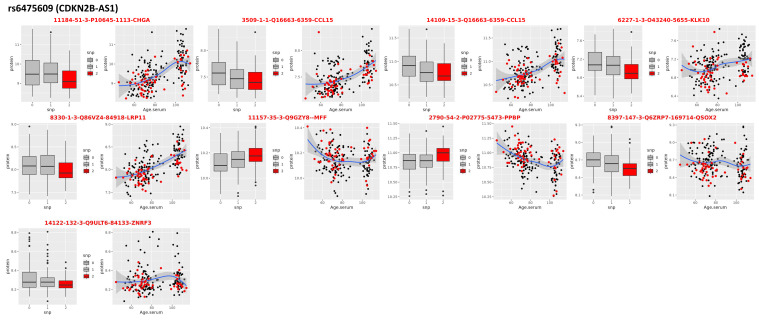
pQTLs in CDKN2B-AS1. Nine aptamers mapping to eight proteins that correlate with genotypes of the SNP rs6475609 in the CDKN2B-AS1 that was associated with extreme human longevity in the discovery GWAS. For each protein: the boxplots on the left show the distribution of the log-transformed protein data by genotype group (red = homozygote genotype for the longevity allele; gray/black = genotypes of carriers of 1 or 2 non-longevity alleles); the scatter plot on the right shows the distribution of the log-transformed protein data (*y*-axis) versus the age of study participants (*x*-axis).

**Figure 4 ijms-24-00116-f004:**
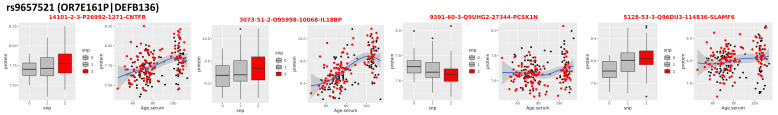
pQTLs in *OR7E161P*|*DEFB136*. Proteins that correlate with genotypes of the SNP rs9657521, which was associated with extreme human longevity in the discovery GWAS. For each protein: the boxplots on the left show the distribution of the log-transformed protein data by genotype group (red = homozygote genotype for the longevity allele; gray/black = genotypes of carriers of one or two non-longevity alleles); the scatter plot on the right shows the distribution of the log-transformed protein data (*y*-axis) versus the ages of study participants (*x*-axis).

**Figure 5 ijms-24-00116-f005:**
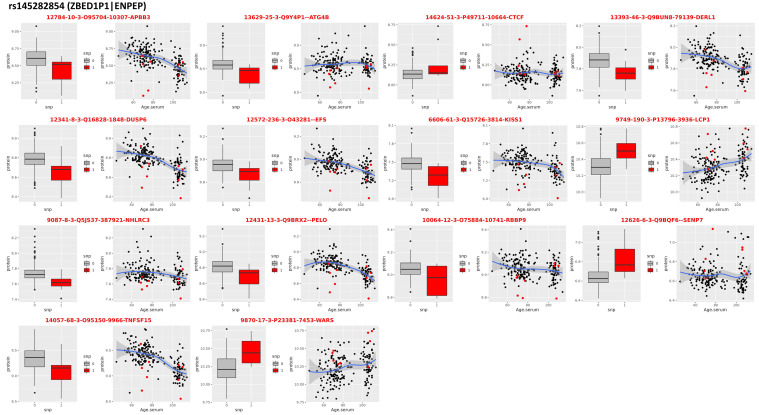
pQTLs in *ZBED1P1*|*ENPEP*. Proteins that correlate with genotypes of the SNP rs145282854 that was associated with extreme human longevity in the discovery GWAS. For each protein: the boxplots on the left show the distribution of the log-transformed protein data by genotype group (red = heterozygote genotype of carriers of the longevity allele; gray/black = genotypes of carriers of two non-longevity alleles); the scatter plot on the right shows the distribution of the log-transformed protein data (*y*-axis) versus the ages of study participants (*x*-axis).

**Table 2 ijms-24-00116-t002:** Protein Signatures for the Top SNPs.

rs6475609 (CDKN2B-AS1)								
SomaScan ID	UniProt ID	Gene	beta	se	t	*p*-Value	FC **	AdjP ***
6227-1_3	O43240	KLK10	−0.09431	0.025082	−3.75987	0.00022	1.244114	0.004196
11157-35_3	Q9GZY8	MFF	0.034337	0.009403	3.651539	0.000328	0.96116	0.112067
3509-1_1	Q16663	CCL15	−0.07418	0.020842	−3.55895	0.00046	1.309531	9.30 × 10^−7^
11184-51_3	P10645	CHGA	−0.20668	0.058159	−3.55364	0.000467	2.041239	7.03 × 10^−7^
8397-147_3	Q6ZRP7	QSOX2	−0.06864	0.019382	−3.54119	0.000488	0.894969	0.075736
14122-132_3	Q9ULT6	ZNRF3	−0.04099	0.01168	−3.50934	0.00055	0.976452	0.424986
14109-15_3	Q16663	CCL15	−0.08507	0.024385	−3.48852	0.000591	1.245471	0.000548
8330-1_3	Q86VZ4	LRP11	−0.07416	0.021678	−3.42106	0.000746	1.361808	2.42 × 10^−9^
2790-54_2	P02775	PPBP	0.06303	0.018485	3.40968	0.000777	0.873098	0.006294
**rs9657521 (OR7E161P|DEFB136)**								
5128-53_3	Q96DU3	SLAMF6	−0.09213	0.025586	−3.60081	0.000395	1.174195	0.006707
3073-51_2	O95998	IL18BP *	−0.08158	0.02414	−3.37965	0.000862	1.313291	1.73 × 10^−8^
9391-60_3	Q9UHG2	PCSK1N *	0.034867	0.010381	3.358835	0.000929	1.044992	0.163917
14101-2_3	P26992	CNTFR *	−0.05845	0.017439	−3.35168	0.000949	1.149051	3.33 × 10^−5^
**rs145282854 (ZBED1P1|ENPEP)**								
12626-6_3	Q9BQF6	SENP7	0.185977	0.044871	4.144741	4.93 × 10^−5^	0.973749	0.619881
12341-8_3	Q16828	DUSP6	−0.11968	0.030611	−3.90953	0.000124	0.905893	4.91 × 10^−7^
12431-13_3	Q9BRX2	PELO	−0.12712	0.032736	−3.88324	0.000138	0.899788	5.94 × 10^−6^
6606-61_3	Q15726	KISS1	−0.20633	0.054011	−3.82019	0.000178	0.936344	0.141048
14624-51_3	P49711	CTCF	0.13336	0.035568	3.749403	0.000228	0.992483	0.129235
9870-17_3	P23381	WARS	0.228915	0.061876	3.699553	0.000275	1.092315	0.041444
13629-25_3	Q9Y4P1	ATG4B	−0.23443	0.063475	−3.6932	0.000282	0.975495	0.854651
9749-190_3	P13796	LCP1	0.181824	0.049431	3.678319	0.000297	1.089198	0.040104
14057-68_3	O95150	TNFSF15	−0.22904	0.063327	−3.61671	0.000372	0.780793	1.10 × 10^−9^
12572-236_3	O43281	EFS	−0.08412	0.023752	−3.54161	0.000488	0.933805	5.38 × 10^−5^
12784-10_3	O95704	APBB3	−0.17642	0.049996	−3.52875	0.000511	0.870573	7.23 × 10^−6^
10064-12_3	O75884	RBBP9	−0.10174	0.028931	−3.51656	0.000534	0.994783	0.918297
13393-46_3	Q9BUN8	DERL1	−0.1092	0.031437	−3.4736	0.000622	0.957411	0.010074
9087-8_3	Q5JS37	NHLRC3	−0.13061	0.037984	−3.4386	0.000704	0.928668	0.018714

* Did not reach proteome-wide significance. ** FC: Fold change comparing protein abundance in controls versus centenarians. Note that FC control to centenarian >1 indicates a protein that decreases in centenarians, while FC control to centenarian <1 indicates a protein that increases in centenarians. *** AdjP: Adjusted *p*-value, FC and AdjP are extracted from the analysis reported in: Protein signatures of centenarians and their offspring suggest centenarians age slower than other humans—Sebastiani—2021—Aging Cell—Wiley Online Library.

## Data Availability

The LLFS data are available on dbGaP (dbGaP Study Accession: phs000397.v3.p3). All other data are not publicly available for privacy reasons.

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
