# Peer review of "A Genome-Wide Association Study of 2304 Extreme Longevity Cases Identifies Novel Longevity Variants"

_ijms, 2022, doi:10.3390/ijms24010116_

Round 1

Reviewer 1 Report

Longevity is essentially the obverse of total mortality.  Extreme longevity (EL), defined as surviving beyond the 99thpercentile of age at death for gender and birth cohort, might appear to focus on mortality at extreme ages, but of course, one must first have avoided dying earlier to be “at risk” of EL, so in this sense, death rates at all ages are involved in the phenotype.  

The EL trait thus seems to pose a puzzling difficulty for epidemiologic analysis: if cases are those who survived to this threshold, who are suitable controls?  They must obviously have died at younger ages, but then how can they really be comparable (including with respect to a host of possible non-genetic factors)?  Presumably they are not just censored by having been born recently enough that they haven’t had the opportunity to reach the EL threshold yet.  Alas, this paper does not really adequately describe the source of controls, saying simply “We defined controls as study participants not achieving the above threshold, or study controls”, although some clarification is provided in reference (7), as “Controls were defined as either male study participants who died at age less than 90 years, or female study participants who died at age less than 95 years, or study-specific controls (Table 1).“ (What the “study specific controls” are is not defined in either paper, nor in reference (5).)  They also allude to the problem I mentioned above, writing that “We showed in ref. (26) that choosing controls with age at death too different from the minimum age of cases can introduce a bias in the estimation of the genetic effects.”  

Reading between the lines, I gather that both cases and controls derive from cohorts (in some, family studies) defined by a range of birth years that would have allowed them the opportunity to attain their respective EL threshold, cases having survived beyond that point (irrespective of whether or not they have since died) and controls having died prior to that threshold (but possibly restricted to not having died too much earlier, as noted in their comment above).  Since all controls and some of the cases have died, their DNA samples must have been obtained at enrollment in the cohort. This also raises the issue of left truncation, as they must have survived long enough to have joined the cohort.  That could be a problem if a survival analysis were done, but maybe less so for the logistic regression of the binary case-control comparison that was actually done.  

To get some insight into this issue, I did a simple lifetable calculation comparing cases as either person-years or deaths past age 100 to controls as either person-years or death prior to age 100.  I was able to persuade myself that a comparison of any of these four possible case-control combinations for a genetic risk factor with a constant hazard ratio at all ages provided a valid estimate of the relative risk under the null hypothesis, although all four comparisons provided somewhat biased estimates under the alternative hypothesis.  Since the goal here is hypothesis testing, not estimation, that bias is not an issue, but of course the different strategies would have different power.  

My point is not that the study design and statistical analysis is wrong, only that it could be described better.  The “replication” cohorts also seem to have used a different, parent-offspring, design, based on a censored survival analysis of parents’ age at death on the genotype of the offspring (the enrolled cohort member).  This is mentioned only in passing as one of the limitations in the last paragraph of the Discussion, but not explained in the methods section.

Minor points

Some explanation of the overlap between the UKB and LifeGen cohorts would be helpful.

Chromosome 9 locus: you point out the difference between the UKB-F and UKB-M effect sizes, but couldn’t you also show whether you saw the same sex effect in the discovery cohorts? 

Further down in the paragraph, I may not have understood the analysis of the expression data, but how are we to see the trend in abundances with age?  It seems like what is being reported is the regression of abundance on genotypes.

Chromosome 11 locus: This seems to have just missed your criterion for genomewide significance and should be described as suggestive.  I’m not sure where the 1x10-5 p-value comes from, maybe one of your earlier papers?

Chromosome 8 locus: wasn’t quite replicated in UKB-M (p=.07) according to your criteria, although this is clarified three lines down.

Duncan C. Thomas

Author Response

Please see the attachment named "v2".  V1 may have wrong page/line numbers.

Reviewer 2 Report

In the paper entitled “A Genome-Wide Association Study of 2,304 Extreme Longevity Cases Identifies Novel Longevity Variants” Bae and colleagues utilized a Genome wide association approach to identify common variants associated with human extreme longevity (EL).   General comments

Overall, the hypothesis underpinning the study is valid. However, there are some points to clarify to understand the solidity of the findings.

First, most of the findings reported in this study are not statistically significant (only the known APOE locus reach the GWA statistically significant threshold of 5*10-8) and some of them have been already reported (see chromosome 9). Indeed, this study is built on a previous meta-analysis (Sebastiani et al. 2017) with the inclusion of 234 new EL cases. I would advise a huge caution to report loci which are at suggestive significant level (Pval<5*10-6). The authors utilised a similar approach in Sebastiani et al. 2017 and reported three suggestive loci on chromosomes 2,6 and 16. However, adding 234 new cases the described “loci” were not validated in this study.

I would also advise to downplay some sentences in the results and the discussion sections. For example, in the result section, describing chromosome 11 locus the authors report:” We observed a significant peak on chromosome 11 that harbors 27 rare SNPs in 52 RPLPOP2 with p<1x10-5”. If the Pvalue is not <5*10-8 the peak is not significant.

Similar in the discussion. “We discovered a novel locus on chromosome 8 (rs9657521) that is associated with EL.”. The SNPs did not reach a GWA significant level and therefore I would apply more caution to define the locus on chromosome 8 as novel locus for EL.

In study the authors changed (lowered) the quality controls threshold applied to the SNPs before the GWA analysis. In the previous study (Sebastiani et al. 2017) the authors utilised the common threshold values for excluding SNPs with: a) poorly imputed (quality score <0.9); b) MAF <0.01 and c) HWE in controls p<10-6 . In this study the authors used a lower imputation quality score (imputation quality score <0.7), included very rare SNPs and did not mention any HWE exclusion in the method section.

Some of the results of this study may be the result of lowering the SNP’s QCs. It is a known fact that SNPs with very low allele frequency (MAF < 0.01) result in very bad imputation (even when the quality score is >0.8). This applies in particular to the locus on chromosome 11. There are several checks that need to be reported to sustain the solidity of the findings on chromosome 11. For example, what are the quality score for the “associated” SNPs?

Moreover, the analysis of very rare SNPs (MAF<0.01) should be conducted with appropriate tools/methods. In the method sections the author report:” We used our GWAS pipeline tool developed by Song et al”. The reader (and this reviewer) needs more details on this pipeline, in particular which is the software is based on and how/if this pipeline does include any method to properly analyse SNPs with very low frequency (MAF<0.01).

At the end of the introduction, the authors report “We sought for replication of our discovery results in three publicly available GWASs” and then mention that the three cohorts are: “UK Biobank (UKB) GWAS of father’s age at death and mother’s age at death, and the meta-analysis of parental lifespan from UKB and 26 independent European-heritage population cohorts (UKB+LifeGen)”. I do agree on the choice of the replication cohorts. However, I do not entirely agree that these are 3 independent cohorts. While UKB-F and UKB-M are two independent datasets (although in this case the results have a gender bias), UKB+LifeGen (maybe) already includes both ULB-F and UKB-M and, therefore, it is not a third independent dataset. I am not sure than define the validation cohorts as three independent GWAs cohorts is entirely appropriate (unless it is clearly explained in the methods section why this is not the case).

Finally, the study in particularly because is based on “suggestive” loci would benefit on a more robust secondary analysis. The authors utilised as validation a proteomic analysis. I would suggest implementing a more robust approach and include a colocalization analysis with the proteomics data rather than limit the evidences on a simple association with the top SNPs. I would suggest to try to include some publicly available datasets of eQTL and pQTL to identify putative mechanisms underlying EL. Similar to the approach utilised by Wu et al. (Nat Commun 9, 918 (2018). https://doi.org/10.1038/s41467-018-03371-0).

Minor comments

In Figure1 and Table1 the authors refer to the discovery GWA as “Mega-Analysis” and  “MEGA-ANALYSIS” respectively. The same term is not utilised (or defined) in the main text. Could the authors please use a consistent way of reporting the discovery analysis in both Figure1 and Table1 and explain/utilise the term in the main text.

In the discussion section the authors report:” We also conducted a regional phenome-wide association 165 study (PheWAS)42 search of the associations between this SNP with 778 traits...”. Why the phewas was done only for this SNP and not performed for all the identified SNPs? Why there was no mention of the methods utilised and the results observed in the previous sections.

In the discussion section the authors report:”…as recently supported in Cristina et al. 2018 and others…”. Please use the surname of the author when reporting this reference.

The locus plots of the loci (as suppl figures) would help to see which genes are present in the identified loci.

The methods in supplementary information may be correct but they not really clear to me (in particular the definition of the deltas). Please try to describe it clearer.

Author Response

Please see the attachment named "v2".  V1 may have wrong/outdated page/line numbers.

Reviewer 3 Report

1. The limitations discussed by the authors do not touch on a few notable points that are related to the assumptions or issues using summary statistic data. For example.

a. Bias from adjustments made in GWAS: Since summary data from published GWAS is used and the authors have to accept the adjustments that have been made in those GWAS, commenting on the likely impact of such adjustments is necessitated.

b. Possibilities of non-linear or interactive (effect moderation) effects but the use of summary statistics makes it unlikely to be checked.

2. These identified genetic variants explain how much variance in EL? Can the authors provide some discussion on this matter?

3. Can the authors provide some discussion on how these findings can be translated to non-European ancestries?

4. Did the authors look into any evidence of a possible causal relationship between the identified variants and EL? Can the authors provide results on fine mapping their significant GWAS locus with haplotype blocks and also obtain 99% credible set variants? 

5. In the statistical analysis part the authors refer to their analysis pipeline which they have previously used however, a reader may find it difficult to follow as they have to meander around different articles. Can the authors provide details of their analysis pipeline for both the GWAS and the pQTL analysis as supplementary material?

6. The authors do not provide any clear information on the R packages/codes that are being used to implement the methods. Therefore, it is not clear how researchers and practitioners alike can apply these methods in real applications and reproduce the results. The practical applicability will be significantly improved if the coding used in its implementation is made available (e.g., a web link with clear instructions for the users).

Round 2

Reviewer 1 Report

I enjoyed reading the authors' responses to my previous critique and found this revision quite satisfactory.

One minor question: in the LGP study, I assume by "genetically matched offspring of parents" you mean matched to the EL cases (not the parents).  Are they matched on the first few GWAS principal components or what?  Might that tend to overmatch on EL-related genes too, thereby reducing power (but not causing bias in a proper matched analysis)?  If this matching is on general ancestry to avoid confounding by population stratificatio, then any loss of power may be insignificant.  I ask only to satisfy my curiosity; I leave it to the authors to decide whether any further comment in the paper is warranted.

Reviewer 2 Report

The authors addressed all the comments, and the manuscript is now clearer than the first version. Although I am not fully convict about the value of reporting suggestive loci, I am overall happy with the current version of the manuscript.